# Impact of Treatment on Quality of Life in Oropharyngeal Cancer Survivors: A 3-Year Prospective Study

**DOI:** 10.3390/cancers16152724

**Published:** 2024-07-31

**Authors:** Victoria Nuñez-Vera, Alberto Garcia-Perla-Garcia, Eduardo Gonzalez-Cardero, Francisco Esteban, Pedro Infante-Cossio

**Affiliations:** 1Department of Oral and Maxillofacial Surgery, Virgen del Rocio University Hospital, 41013 Seville, Spain; victorianunezvera@gmail.com (V.N.-V.); agarciaperla@us.es (A.G.-P.-G.); dr.gcardero@gmail.com (E.G.-C.); 2Department of Surgery, School of Medicine, University of Seville, 41009 Seville, Spain; festebano@us.es; 3Department of Otorhinolaryngology, Virgen del Rocio University Hospital, 41013 Seville, Spain

**Keywords:** oropharyngeal cancer, EORTC QLQ-C30, EORTC H&N-35, quality of life, head and neck cancer treatment

## Abstract

**Simple Summary:**

Oropharyngeal carcinoma (OPC) and its treatments can negatively affect patient quality of life (QoL). This study focused on assessing the impact on QoL in OPC survivors using the EORTC QLQ-C30 and EORTC QLQ-H&N35 scales before treatment and in the first and third years after treatment. Of 72 patients treated for OPC, 51 completed all questionnaires over 3 years. A variable deterioration of the QoL scores was detected prior to treatment. Most items of QoL worsened significantly after treatment and during the first year and improved in the third year. Advanced-stage cancer and definitive chemoradiotherapy treatment showed the worst scores. Patients treated with an open surgical approach exhibited significant deterioration compared to transoral surgery. This prospective long-term study in a homogeneous group of OPC survivors showed that although QoL was generally good, patients needed a long period of time to recover from both cancer and treatment effects.

**Abstract:**

(1) Background: This prospective study aimed to assess the impact on quality of life (QoL) from pretreatment to 3 years after treatment in oropharyngeal carcinoma (OPC) survivors. (2) Methods: QoL was measured with the EORTC QLQ-C30 and EORTC QLQ-H&N35 scales before treatment and in the first and third years. (3) Results: Of 72 patients, 51 completed all questionnaires over 3 years. A variable deterioration of QoL scores was detected before treatment. Most items worsened significantly after treatment and during the first year and improved in the third year. Advanced-stage cancer and definitive chemoradiotherapy treatment showed the worst scores. At 3 years, patients who underwent surgery with adjuvant radiation therapy/chemotherapy had significantly better scores on global QoL and emotional functioning compared to those treated with definitive chemoradiotherapy, who also reported problems with sticky salivation and dry mouth. Patients treated with an open surgical approach showed significantly greater deterioration in physical and role functioning compared to transoral surgery. (4) Conclusions: This long-term prospective study is the first in Spain to use EORCT scales in a homogeneous group of OPC survivors. QoL was generally good, although patients needed a long period of time to recover from both cancer and side effects of treatment. Advanced-stage cancer and definitive chemoradiotherapy showed the worst scores.

## 1. Introduction

Oropharyngeal cancer (OPC) causes a significant impact on the quality of life (QoL) of surviving patients because it impairs essential functions such as swallowing, chewing, salivation, and speech. This impact is intensified by aggressive treatments such as radical surgery and radiotherapy/chemotherapy (RT/CT) that generate alterations that impair the patient’s functionality and QoL. Systematic reviews on QoL in OPC have reported significant problems and side effects of treatment that persist up to 1 year after diagnosis [1,2]. Although initially the patient’s main concern is their survival, even assuming a possible deficit secondary to therapy, their focus will later be on maintaining and improving their QoL to cope with the impact of cancer and its treatment.

In recent decades, the incidence and mortality for OPC have experienced a worrying increase worldwide [3]. Overall, neoplasms originating in the oral cavity, oropharynx, hypopharynx and larynx constitute the ninth leading cause of death worldwide [4]. These data highlight the need to pay special attention to the QoL of patients. The most recognised predictors of QoL in head and neck cancer (HNC) include specific tumour location, stage of disease and type of treatment [5,6]. But regardless of survival, recurrence and disease-free years, addressing QoL will not only aid therapeutic decision-making in clinical practise and early detection of potential side effects, but its improvement may even correlate with survival rates [7].

Several tools have been implemented to measure QoL. The European Organisation for Research and Treatment of Cancer Quality of Life Questionnaire (EORCT QLQ-C30) and its specific module for head and neck cancer (EORCT H&N-35) have become some of the most widely used reference instruments worldwide [8,9,10,11]. However, longitudinal studies of QoL in patients treated for HNC with more than 12 months of follow-up considering different treatment modalities are scarce [1,2] and have shown that, although treatment may be effective in disease control, it has important consequences with persistent symptoms such as dysphagia, xerostomia, and psychosocial problems. These previous studies have reported a deterioration of QoL after treatment and improvement at 6 months and during the first year, although many items will not reach baseline values.

This prospective study focused on analysing the long-term QoL of patients treated for OPC using the EORTC QLQ-C30 and EORTC H&N-35 scales at three control time points: pretreatment and 12 months and 36 months post-treatment. The aim was to calculate clinically significant changes (≥10) in QoL for each follow-up checkpoint considering tumour stage (I/II vs. III/IV) and treatment modality (surgery alone, surgery with adjuvant RT+/−CT or definitive CT-RT).

## 2. Materials and Methods

### 2.1. Study Design

This single-centre, prospective, longitudinal clinical study assessed QoL three years after treatment in surviving patients diagnosed with oropharyngeal squamous cell carcinoma at the Virgen del Rocio University Hospital, Seville, Spain. The study period covered from 2007 to 2017. Patient data were collected from the departments of Oral and Maxillofacial Surgery, Otorhinolaryngology and Radiation Medical Oncology. The study adhered to ethical guidelines and obtained approval from the Hospital Ethics Committee. All patients involved gave their informed consent to participate.

### 2.2. Participants

The inclusion criteria were patients with squamous-type OPC who were over 18 years of age, treated with curative intent, and able to understand and respond to the QoL questionnaires. This study used the International Classification of Diseases (ICD) coding systems (ICD-9 and ICD-10) to categorise tumour locations. These locations included the base of the tongue (C01), the lingual tonsil (C02.4), the soft palate (C05.1), and the tonsil/oropharynx (146, C09-10) (Figure 1). In addition, tumour staging followed the seventh edition of the American Joint Committee on Cancer (AJCC) guidelines, with classifications of early (stages I–II) or advanced (stages III–IV) [12]. Patients with malignant neoplasms of the nasopharynx, hypopharynx, and salivary glands; non-squamous cell cancers; and recurrences, distant metastases, or second primary tumours in the maxillofacial area that did not receive treatment with curative intent or did not respond to questionnaires were excluded.

### 2.3. Study Procedure

At the first visit, patients eligible for this study were identified. If they were interested in participating, the main researcher explained the procedure and signed the consent form. Upon diagnosis and prior to starting treatment, a comprehensive assessment of the patient was carried out. This assessment included sociodemographic information (age, gender, educational level, employment status, and marital status) and smoking and alcohol consumption habits. Functional status was measured using the Karnofsky scale. Cancer data (tumour location, size, and staging) were evaluated. Additionally, data on comorbidities, including cardiovascular, respiratory, gastrointestinal, renal, endocrine, neurological, rheumatological, and immunological diseases, were collected. Previous diagnoses of any tumours were also documented.

Treatment was planned by the Multidisciplinary HNC Committee of our hospital based on TNM stage and comorbidities, taking into account the patient’s preferences. Our treatment scheme followed the guidelines of international protocols [13]. Most early-stage carcinomas (I–II) were treated with surgery (+/−adjuvant RT), while most advanced-stage carcinomas (III–IV) underwent a multimodality approach including surgery with adjuvant RT+/−CT or concomitant definitive CT-RT with organ preservation.

Surgery: Radical excision was performed with a safety margin, with or without adjuvant RT. Indications for adjuvant RT included tumour size and grade of differentiation, deep invasion, nerve invasion, regional lymph node involvement with more than one positive lymph node, extranodal tumour spread, perineural invasion, lymphovascular invasion, and narrow surgical margins. All surgically treated patients underwent at least a selective neck dissection. Depending on the tumour stage, ipsilateral or bilateral cervical dissection was performed. An open surgical approach was performed via an endobuccal/cervical (cheilo/mandibulotomy) or transoral approach (Figure 2 and Figure 3). Local, regional or free flaps were used for reconstruction, depending on the case.

Definitive RT: A total of 70 Gy was administered to the primary lesion and 50 Gy to the nodal levels (2 Gy/day fraction). The modality consisted of 3D conformal RT or IMRT (Figure 4).

CT: As first line, 100 mg/m^2^ of cisplatin every 21 days × 3 cycles, and as second line, cetuximab 400 mg/m^2^ prior to RT as loading dose and 250 mg/m^2^ concomitantly with weekly RT for 6 doses.

The recruitment process is shown in Figure 5. Of the 76 patients who met the inclusion criteria, 4 were initially excluded because they refused to participate. At 36 months, 51 surviving patients completed the study.

### 2.4. Assessment of QoL

In this study, the EORTC QLQ-C30 and the EORTC QLQ-H&N35 Spanish-validated questionnaires were used to assess patients’ QoL [14]. Baseline measurements were collected before the start of treatment, followed by additional assessments at 12 and 36 months. A researcher facilitated the administration of the questionnaires prior to treatment and intervened only to clarify any doubts that patients might have, without influencing their responses.

The EORTC QLQ-C30 questionnaire assesses the general impact of cancer and its treatment. It comprises: 5 Functional Scales: Physical, Role, Emotional, Cognitive, and Social functioning (higher scores indicate better functioning), 3 Symptom Scales: Fatigue, Nausea/Vomiting, and Pain (higher scores indicate greater symptom severity), 6 Individual Items: Dyspnoea, Insomnia, Appetite Loss, Constipation, Diarrhoea, and Financial Difficulties (higher scores indicate greater problems), and 1 Global Health and QoL Scale (higher scores indicate better overall health and QoL). The specific EORTC QLQ-H&N35 questionnaire focuses on HNC-related QoL and is used in conjunction with the QLQ-C30. It includes: 7 Symptom Scales: Pain, Swallowing, Sense, Speech, Social Eating, Social Contact, and Sexuality (higher scores indicate greater symptom severity), 11 Unique Items: Teeth, Opening Mouth, Dry Mouth, Sticky Saliva, Coughing, Feeling Ill, Pain Killers, Nutritional Supplements, Feeding Tube, Weight Loss, and Weight Gain (higher scores indicate greater problems). Interpretation of both questionnaires followed the established EORTC scoring guidelines.

### 2.5. Statistical Analyses

Statistical analysis was accomplished with SPSS version 22.0 for Windows (SPSS Inc., Chicago, IL, USA). Descriptive analysis was performed by analysing the absolute and relative frequencies (percentages) of qualitative variables. For quantitative variables, the median (interquartile range) was used, depending on whether or not they followed a normal distribution (verified by the Kolmogorov–Smirnov test). To compare the different quantitative variables, the Mann–Whitney U test or the Kruskal–Wallis test was used when these variables did not have a normal distribution. Categorical variables were analysed using the chi-square test or Fisher’s exact test, when necessary. For continuous variables, the Friedman test was used to compare measurements at different time points (baseline, 12 months and 36 months).

Statistical analyses were performed on the 51 patients who completed the questionnaire at the three checkpoints, excluding those who did not complete the study due to death. For *p*-values, a statistical significance level of 5% (*p* ≤ 0.05) was chosen. When the Bonferroni correction was applied to analyse intragroup variations between the three checkpoints, the statistical significance was *p* ≤ 0.017. Raw scores from the QoL surveys were transformed into a linear scale from 0 to 100 to standardise them. The study population was divided into subgroups according to tumour stage (early, advanced) and treatment (surgery, RT, CT, types of surgical approaches). Changes (Δ) in absolute scores were interpreted as follows: (1) an increase in score (positive value) is an improvement in QoL, and (2) a decrease in score (negative value) is a deterioration of QoL. To calculate the minimal clinically important difference (MCID), a change in absolute scores on the scales of ≥10 points over time was considered [15,16,17]. A clinically significant change was defined as a score difference large enough to have implications for patient treatment or care [18].

## 3. Results

### 3.1. Characteristics of Patients

The main characteristics of the study group are presented in Table 1. The 72 patients who participated in the study were predominantly male (84.7%). The mean age was 59.1 years (range 28–85 years). The mean age in women was 65.4 years and that in men was 57.8 years. The most frequent medical history included hypertension (37.5%) and diabetes (19.45%). Of the patients, 61.5% were smokers, 13.5% had quit, and 25% had never smoked, while 52.7% were alcohol drinkers, 4.2% were ex-drinkers, and 43.1% had never drunk alcohol. The predominant tumour site was the soft palate (86.1%). A total of 58.3% of patients were classified as advanced stage III-IV, with a higher percentage in women (72.2%) than in men (55.7%). Most patients (87.5%) received surgical treatment with neck dissection and 61.1% received postoperative RT/CT. In 81% (51/63) of the cases, it was possible to perform a complete surgical resection of the tumour with a safe margin. Concomitant definitive RT-CT was indicated in 12.5%. Some type of flap reconstruction (local, pedicled, or free) was performed in 62.5%.

### 3.2. QoL According to Cancer Stage

The analysis at the three checkpoints according to diagnostic stage (early, advanced) was calculated using only the data of those patients who completed the study (Table 2 and Table 3). The intergroup comparative analysis with the QLQ C-30 showed significant differences (*p* ≤ 0.05) in 6 parameters at baseline and in 2 parameters at 12 months, and with the QLQ H&N-35, in 10 parameters at baseline, in 14 parameters at 12 months, and in 7 parameters at 36 months.

Intragroup analysis of changes over time showed significant statistical differences (*p* ≤ 0.017) with MCID ≥ 10 in 4/15 parameters with the QLQ-C30. We found deterioration at 12 months with improvement at 36 months in financial difficulties and global health status (early and advanced stage), role functioning (early stage) and fatigue (advanced stage). Regarding the QLQ H&N-35, statistically significant differences (*p* ≤ 0.017) were found with MCID ≥ 10 in 2/18 parameters. A deterioration was found at 12 months with an improvement at 36 months in mouth opening problems and weight loss (early and advanced stage).

Figure 6 represents the global health status item of the QLQ-C30. In both early- and advanced-stage patients, a significant score reduction was observed at 12 months with recovery at 36 months, reaching values higher than those at baseline. However, the deterioration at 12 months and its improvement at 36 months was more marked in the advanced stage.

### 3.3. QoL According to Cancer Treatment

The analysis at the three checkpoints according to the treatment performed (surgery+/−CT-RT, definitive CT-RT) was calculated using only the data of those patients who completed the study (Table 4 and Table 5). The intergroup comparative analysis with the QLQ C-30 showed significant differences (*p* ≤ 0.05) in one parameter at baseline and in two parameters at 36 months, and with the QLQ H&N-35, in two parameters at baseline, in one parameter at 12 months and in two parameters at 36 months.

Intragroup analysis of changes over time showed significant statistical differences (*p* ≤ 0.017) with MCID ≥ 10 in 1/15 parameters with the QLQ-C30. Deterioration was found at 12 months with an improvement at 36 months in fatigue (surgery+/−RT-CT). Regarding the QLQ H&N-35, there were statistically significant differences (*p* ≤ 0.017) with MCID ≥ 10 in 0/18 parameters.

Figure 7 represents the global health status item of the QLQ-C30. Both in patients who received surgery+/CT-RT and in those treated with concomitant definitive CT-RT, a reduction in score at 12 months and recovery at 36 months with values higher than baseline control were observed.

### 3.4. QoL According to the Type of Surgical Approach

The analysis at the three checkpoints according to the type of surgical approach (open surgery, transoral) was calculated using only the data of those patients who completed the study (Table 6 and Table 7). The intergroup comparative analysis with the QLQ C-30 showed significant differences (*p* ≤ 0.05) in one parameter at baseline, in two parameters at 12 months, and in two parameters at 36 months, and with the QLQ H&N-35, in nine parameters at baseline, in nine parameters at 12 months and in five parameters at 36 months.

Intragroup analysis of changes over time showed significant statistical differences (*p* ≤ 0.017) with MCID ≥ 10 in 6/15 parameters with the QLQ-C30. We found deterioration at 12 months with improvement at 36 months in physical and emotional function (open or transoral surgery) and fatigue (open surgery) and improvement at 12 months and 36 months in global health status (open or transoral surgery). Regarding the QLQ H&N-35, there were statistically significant differences (*p* ≤ 0.017) with MCID ≥ 10 in 2/18 parameters. A deterioration was found at 12 months with an improvement at 36 months in pain (open or transoral surgery) and problems in mouth opening (open surgery).

Figure 8 represents the global health status item of the QLQ-C30. Significant differences were observed in both patients undergoing open and transoral surgery with a low baseline score, which increased in the first year and even more at 36 months.

## 4. Discussion

Our study focused on a prospective evaluation of the QoL of patients treated for OPC using the EORTC QLQ-C30 and QLQ-H&N35 scales at three time points, pretreatment and 12 months and 36 months post-treatment. A longitudinal intergroup and intragroup comparative analysis was performed, considering tumour stage and different types of treatment and surgical approaches. Our results showed significant variations in multiple QoL parameters. Before treatment, most of the patients reported a variable degree of deterioration in several specific aspects. At 12 months, an increase in symptoms was evident, while at 36 months, scores on most QoL items stabilised, reaching values similar to baseline and, in some cases, even improved. The findings of our study revealed three patterns of behaviour with the QLQ-C30: (1) deterioration at 12 months and improvement at 36 months; (2) deterioration at 12 and 36 months; and (3) improvement at 12 and 36 months. The QLQ-H&N35 scale reflected similar patterns of QoL comportment, although it was more sensitive and specific in detecting a greater number of clinically altered items in the intergroup analysis compared to the generic QLQ-C30, indicating that this specific questionnaire is relevant in the evaluation of patients with OPC.

The results of our study are broadly consistent with most published studies on QoL in patients with HNC [19]. The review by Rathod et al. [20] analysed the QoL in patients undergoing surgery with or without adjuvant RT-CT; the most commonly used specific instruments were the UWQoL (University of Washington QoL Questionnaire) and the EORTC QLQ-H&N35, while the most commonly used general instrument was the EORTC QLQ-C30. In general, patients who survive OPC have lower QoL values compared to those with oral cavity cancer [6,21], possibly because they are exposed to more aggressive treatments in an anatomical area where functional impairment is greater, and because they are treated in more advanced stages. In the review conducted by Ghazali et al. [22] in patients with HNC, it was considered that there is a change in the trend of QoL studies. There is a current preference for the use of more specific questionnaires to assess impairments in specific areas that affect activities of daily living, such as dysphagia, xerostomia, and oral mucositis. Early detection and treatment of these symptoms are crucial to improving QoL. At the same time, the importance of conducting studies that take into account the perspectives of HNC survivors are emphasised to implement comprehensive plans that address their long-term needs.

Interpretation of reported data on QoL in OPC is limited by multiple confounding factors, such as the use of different questionnaires, small samples with heterogeneous populations, short follow-up, and loss of patients. Furthermore, treatment modalities are not equally balanced between studies and by tumour stages. In our review of the literature, nine longitudinal studies using the EORTC QLQ-30 and EORTC H&N-35 instruments were found in patients with OPC [6,7,23,24,25,26,27,28,29]. In general, all of these studies using EORTC questionnaires indicate that the QoL of patients with OPC tends to worsen during the first year after treatment due to acute side effects and patient adaptation to their new condition. However, after the second and third years, the QoL usually improves and stabilises. This depends on several factors, such as tumour characteristics (staging) and the treatment performed (surgery, RT, CT). However, studies published to date show remarkable heterogeneity depending on treatment modality (surgery, RT, and CT, alone or combined) and individual factors (age, sex, comorbidities, social and psychological support) [6,30,31].

We divided the study population into subgroups of patients according to tumour stage and treatment provided to try to identify which impairments patients had and which most affected their activities of daily living after treatment and will therefore require additional specific support for recovery [32]. Generally speaking, tumour staging was significantly and clinically associated with aspects related to QoL, to the extent that patients with early-stage tumours showed better scores both initially and in the long term compared to more advanced stages. QLQ-H&N35 was more sensitive at detecting changes between both groups, but with worse scores in advanced stages. In the third year, a clinically significant worsening of swallowing, speech, social eating, and social contact problems was found. In advanced stages, any treatment will result in a significant deterioration of QoL.

In the review conducted by Roets et al. [2], worse QoL related to surgical approaches was reported. In our study, patients undergoing surgery with adjuvant RT+/−CT showed greater physical deterioration, worse physical, role, and social function, and increased fatigue one year after treatment. The observed differences may be due both to the effects of treatment and the fact that patients who received adjuvant RT had the lowest scores at the beginning of the study. Side effects of adjuvant RT, such as residual radiodermatitis and mucositis, constriction of the masticatory muscles, dental deterioration, and xerostomia, resulted in worse scores on items such as swallowing, pain, dry mouth, sticky saliva, reduced mouth opening, sense problems, speech problems, and social eating problems. The adverse effects of RT tend to decrease after a year, remaining as residual effects on a case-by-case basis. In contrast, patients who had only surgery reported better overall scores in terms of activity, salivation, taste, and postoperative pain.

Following the popularisation of minimally invasive approaches and techniques, there has been renewed interest in the primary surgical treatment of OPC, as these procedures carry less morbidity compared to open surgery [28,33]. In our study, we highlighted that those patients who underwent open surgery had greater alterations in physical and role functioning compared to those who underwent transoral surgery. The same is true for symptoms such as speech problems and dry mouth 36 months after treatment. Overall, transoral surgery procedures scored better on symptom scales than open surgery procedures. In many centres, RT alone has traditionally been the treatment of choice, but with the advent of transoral surgery and TORS (transoral robotic surgery), a higher QoL has been shown to be achieved [34]. In any case, more advanced-stage tumours require more aggressive surgical procedures that are associated with a lower QoL and increased morbidity [13].

Our study included 72 patients, 51 of whom survived for three years. Twenty-one patients died of cancer, representing a survival rate of 70.83%. A strength of our study was that all survivors completed the questionnaires at all three checkpoints, as the participants were very committed to the care of their disease and attended their follow-up appointments on time. Data were collected prospectively, which minimised recall bias and patient loss. However, this suggested a possible risk of positive selection bias, as the QoL of survivors could be overestimated by not considering data from the deceased, as presumably the scores would have been worse. Longitudinal follow-up allowed us to observe changes and trends over time.

Our results follow the trend in the data reported in the meta-analysis by Høxbroe Michaelsen et al. [1], revealing that more than 85% of patients were in an advanced stage at the time of initial presentation, with moderate to large clinically important changes in the domains of dry mouth, sticky saliva, dysphagia, and eating problems. In our study, advanced-stage patients constituted 58.3%. Regarding the treatment performed, the meta-analysis by Høxbroe Michaelsen et al. [1] highlights that approximately 50% of patients received surgical treatment and 50% received non-surgical treatments. In our study, 87.5% underwent surgical treatments with adjuvant RT or RT-CT (61.1%). It is reasonable to deduce that this high percentage of surgical treatments was due to the fact that most of our patients had lesions in the most anterior part of the oropharynx and in earlier stages, which were therefore more accessible and resectable by surgery. Specifically, 86.12% of the lesions were located in the soft palate, and the rest affected more posterior areas (tonsils, base of the tongue, or lateral wall of the pharynx).

Regarding the item “global health status” of QLQ C-30 at 36 months, it should be noted that in all subgroups, the surviving patients exceeded values of 80 points (Figure 6, Figure 7 and Figure 8). Despite low scores at 12 months in this item, a large improvement was observed at 36 months. A significant difference (*p* = 0.001) was observed in the subgroup undergoing surgery (+/−RT-CT) at 36 months compared to the group receiving concomitant definitive RT-CT. However, it should be noted that due to the small number of patients in the latter group, the results should be interpreted with caution, although they are in line with most studies in the literature reporting that long-term surgical treatments result in better QoL than RT-CT treatments [35,36].

We acknowledge several notable limitations in our study. As this was a prospective cohort study, participants were not randomly assigned to treatment groups to compare different therapeutic modalities that could only have been evaluated with a randomised study. Second, the treatment modalities were unbalanced between the subgroups, as more advanced cancers are often treated with more aggressive treatments, which could lead to worse QoL outcomes. Third, patients face significant emotional stress and time constraints related to diagnosis and multiple visits that may condition their ability to participate in research and affect their perception of functioning. Fourth, the relatively small sample sizes within each treatment, particularly the CT-RT group, restrict the generalizability of our findings and increase the possibility of response bias. Finally, we did not collect information on HPV status since its determination was not performed routinely in our hospital during the period of our study and its presence did not change our therapeutic approach [37,38]. It is well known that HPV-associated OPC has a better prognosis [39,40,41] and some authors have described a better QoL in HPV-negative OPC [28,41,42]. However, it should be noted that in our series, there was a high percentage of smokers/ex-smokers and drinkers/ex-drinkers, which leads us to hypothesise that the role of HPV was biased and therefore might have been irrelevant in our series. Consequently, regardless of HPV status, we assumed that tumour stage and treatment modality were the main determinants of patient QoL.

## 5. Conclusions

This long-term prospective study is the first in Spain to use the EORCT scales in a homogeneous group of OPC survivors. The QoL of survivors was generally good, although they needed a long period of time to recover from both the disease and the effects of treatment. Our study found that most of the QoL items worsened significantly after treatment and improved from the first to the third year. At 3 years, few parameters had not returned to pretreatment values. In the 3-year longitudinal analysis, patients undergoing surgery with or without adjuvant RT+/−CT had significantly better scores in QoL and emotional functioning on QLQ-C30 compared to those treated with definitive CT-RT. The QLQ-H&N35 is more specific because it detects more parameters with significant clinical changes, especially the problems of sticky saliva and dry mouth, difficulty in mouth opening, and pain.

From the results of our study, it can be concluded that tumour staging and treatment play an important role in the perception of the QoL of patients with OPC. Patients with more advanced-stage tumours and treated with RT+/−CT presented the worst outcomes. In addition, those who underwent open surgery showed significantly greater deterioration in physical and role functioning compared to transoral surgery.

## Figures and Tables

**Figure 1 cancers-16-02724-f001:**
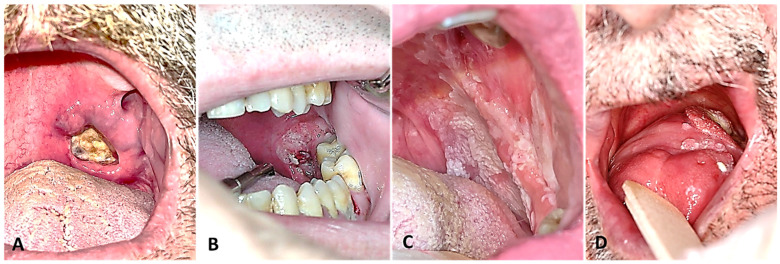
Locations of OPC: (**A**) soft palate, (**B**) tonsillar fossa, (**C**) lateral pharyngeal wall, and (**D**) base of the tongue.

**Figure 2 cancers-16-02724-f002:**
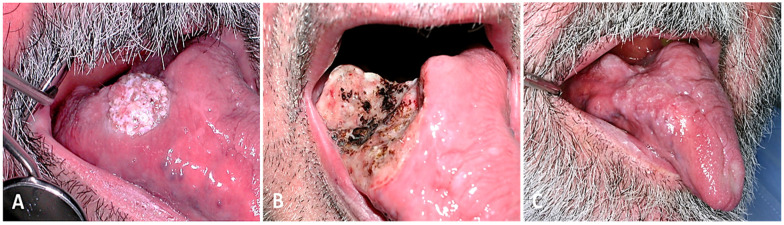
Exophytic OPC at the base of the tongue. CO_2_ laser excision. (**A**) Preoperative view, (**B**) postoperative view in the first week, and (**C**) postoperative view in the second month.

**Figure 3 cancers-16-02724-f003:**
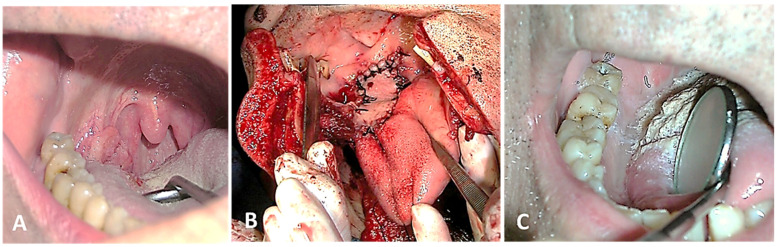
OPC in the tonsil. Excision by open approach with cheilomandibulotomy and reconstruction with pedicled flap on the superior thyroid artery. (**A**) Preoperative, (**B**) intraoperative, and (**C**) postoperative view.

**Figure 4 cancers-16-02724-f004:**
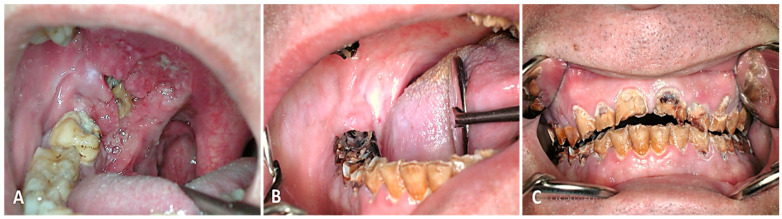
OPC in the anterior tonsillar pillar and soft palate treated with concomitant RT (70 Gy) and CT (3 cycles of cisplatin). (**A**) Pretreatment view and (**B**,**C**) result after 3 years.

**Figure 5 cancers-16-02724-f005:**
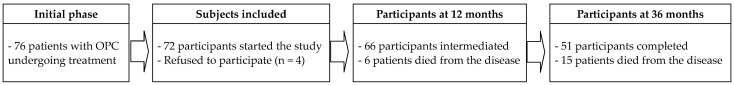
Recruitment of patients at the study checkpoints.

**Figure 6 cancers-16-02724-f006:**
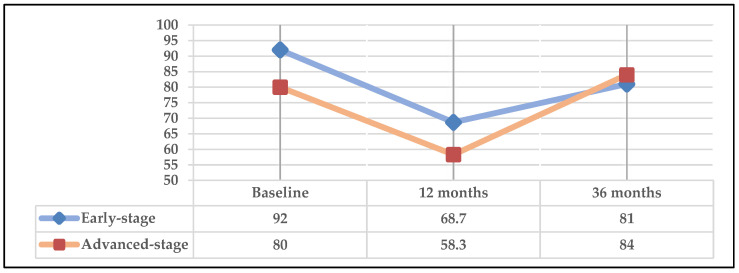
Global health scale according to cancer stage.

**Figure 7 cancers-16-02724-f007:**
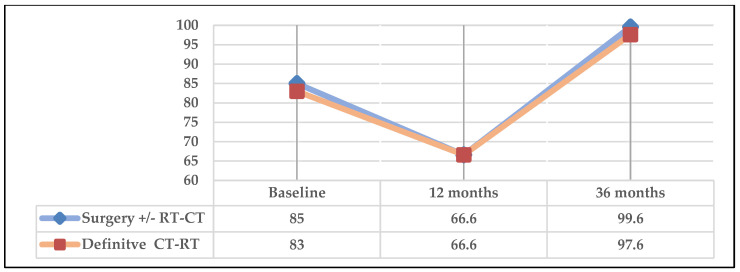
Global health scale according to cancer treatment.

**Figure 8 cancers-16-02724-f008:**
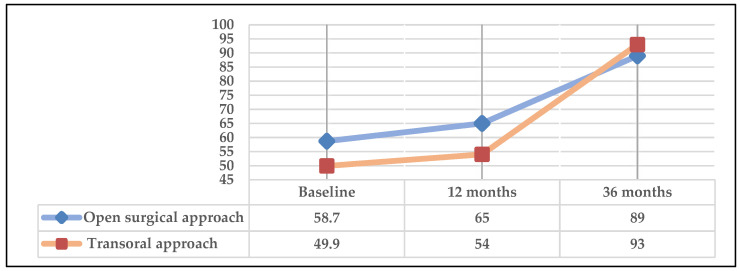
Global health scale according to the type of surgical approach.

**Table 1 cancers-16-02724-t001:** Characteristics of the 72 patients included in the study.

	All	Male	Female
Gender, n (%)	72 (100)	61 (84.7)	11 (15.3)
Age, M (SD)	59.1 (11)	57.8 (10)	65.4 (13)
Comorbidity			
Yes, n (%) No, n (%)	32 (44.5) 40 (55.5)	27 (44.3) 34 (55.7)	5 (45.5) 6 (54.5)
Primary subsite			
Soft palate, n (%) Tonsil/tongue base/pharyngeal wall, n (%)	62 (86.1) 10 (13.9)	54 (88.5) 7 (11.5)	8 (72.7) 3 (27.3)
Tumour stage			
T1, n (%) T2, n (%) T3, n (%) T4, n (%)	18 (25) 24 (33.3) 11 (15.3) 19 (26.4)	13 (21.3) 22 (36.1) 9 (14.8) 17 (27.8)	5 (45.5) 2 (18.2) 2 (18.2) 2 (18.2)
Nodal stage (clinical)			
c0, n (%) cN1, n (%) cN2, n (%) cN3, n (%)	37 (51.4) 26 (36.1) 6 (8.3) 3 (4.2)	33 (54.1) 23 (37.7) 4 (6.6) 1 (1.6)	4 (36.4) 3 (27.3) 2 (18.2) 2 (18.2)
Stage AJCC			
Early (I/II), n (%) Advanced (III/IV), n (%)	30 (41.7) 42 (58.3)	27 (44.3) 34 (55.7)	3 (27.3) 8 (72.7)
Treatment modality			
Surgery, n (%) Surgery + RT, n (%) Surgery + RT-CT, n (%) Definitive concomitant CT-RT, n (%)	19 (26.4) 38 (52.8) 6 (8.3) 9 (12.5)	17 (27.9) 33 (54.1) 5 (8.2) 6 (9.8)	2 (18.2) 5 (45.5) 1 (9.1) 3 (27.3)
Neck dissection			
Yes, n (%) No, n (%)	66 (91.7) 6 (8.3)	57 (93.4) 4 (6.6)	9 (81.8) 2 (18.2)
Flap reconstruction, n (%)	45 (62.5)	40 (65.6)	5 (45.5)
Tracheostomy, n (%)	10 (13.9)	8 (13.1)	2 (18.2)

**Table 2 cancers-16-02724-t002:** QLQ-C-30 scores according to cancer stage (early, advanced) at baseline, 12 months and 36 months.

	Baseline P_50_ [P_25_–P_75_]			12 Months P_50_ [P_25_–P_75_]			36 Months P_50_ [P_25_–P_75_]								
	Early (N = 30)	Advanced (N = 42)	*p* (1)	Early (N = 26)	Advanced (N = 40)	*p* (1)	Early (N = 19)	Advanced (N = 32)	*p* (1)	*p* (2)	*p* (3)	Δ1	Δ2	Δ3	Δ4
**QLQ-C-30 Functional scales (100 = favourable)**															
Global health status Physical functioning Role functioning Emotional functioning Cognitive functioning Social functioning	92 [75–100] 98 [90–100] 99 [90–100] 75 [60–80] 98 [90–100] 99 [90–100]	80 [58–90] 98 [90–100] 83 [80–99] 70 [65–75] 98 [90–100] 83 [80–99]	**0.002** 0.222 **0.001** 0.540 0.901 **0.007**	68.7 [55–86] 95.8 [90.2–100] 79.8 [66–89.5] 80.2 [78–89.2] 99.8 [97–100] 79.8 [73–94.6]	58.3 [24–65.9] 87.3 [83.2–100] 82.8 [66–88.4] 84.2 [74–90.6] 99.8 [83.2–100] 80.8 [73–94.2]	**0.002** 0.220 0.156 0.154 0.190 0.170	81 [74–86.6] 95.8 [90.2–100] 99.8 [95.7–100] 90.2 [89.2–100] 99.8 [96.5–100] 99.8 [96.5–100]	84 [74–86.6] 93.3 [90.2–100] 99.8 [95.7–100] 91.8 [90.1–100] 99.8 [96.5–100] 99.8 [96.5–100]	0.839 0.220 0.156 0.154 0.190 0.367	**0.003** **0.009** **0.003** 0.033 0.567 0.025	**0.012** **0.005** **0.012** **0.007** 0.785 **0.011**	−23.3 −2.2 −19.2 +5.2 +1.8 −19.2	+12.3 0 +20 +10 0 +20	−21.7 −10.7 −0.2 +14.2 +1.8 −2.2	+25.7 +6 +17 +7.6 0 +19
**QLQ-C30 Symptom scales (100 = unfavourable)**															
Fatigue Nausea and vomiting Pain Dyspnoea Insomnia Appetite loss Constipation Diarrhoea Financial difficulties	0 [0–0] 0 [0–0] 18 [10–20] 0 [0–0] 0 [0–0] 0 [0–0] 0 [0–0] 0 [0–0] 0 [0–0]	11 [0–15] 0 [0–0] 45 [35–58] 0 [0–0] 0 [0–0] 5 [0–8] 0 [0–0] 0 [0–0] 0 [0–0]	**0.003** 0.904 **0.003** 0.808 0.564 **0.001** 0.898 0.104 0.886	0 [0–0] 0 [0–0] 16.6 [6.6–33.3] 0 [0–0] 0 [0–0] 0 [0–0] 0 [0–0] 0 [0–0] 25.5 [15–50]	22 [0–30] 0 [0–0] 26.6 [4.1–36.6] 0 [0–0] 0 [0–0] 0 [0–0] 0 [0–0] 0 [0–0] 48.3 [20–56.6]	**0.001** 0.190 0.220 0.190 0.156 0.468 0.189 0.110 0.088	6.8 [0–11.1] 0 [0–0] 6.6 [0–15.2] 0 [0–0] 0 [0–0] 0 [0–0] 0 [0–0] 0 [0–0] 0 [0–0]	9.9 [0–16.6] 0 [0–0] 0 [0–0] 0 [0–0] 0 [0–0] 0 [0–0] 0 [0–0] 0 [0–0] 0 [0–0]	0.222 0.190 0.220 0.190 0.156 0.222 0.189 0.110 0.268	**0.006** 0.673 **0.005** 0.457 0.578 0.579 0.564 0.578 0.023	**0.005** 0.842 0.033 0.478 0.479 0.348 0.567 0.454 **0.001**	0 0 +1.4 0 0 0 0 0 −25.5	−6.8 0 +10 0 0 0 0 0 +25.5	−11 0 +18.4 0 0 +5 0 0 −48.3	+12.1 0 +26.6 0 0 0 0 0 +48.3

Abbreviations: P_50_ = Median score. [ ] = Interquartile range. Significance *p*-value (1) = Intergroup comparative analysis at each checkpoint (results in bold considered significant *p* ≤ 0.050). Significance *p*-value (2) = Intragroup comparative analysis according to early stage between the three checkpoints (results in bold considered significant *p* ≤ 0.017). Significance *p*-value (3) = Intragroup comparative analysis according to advanced stage between the three checkpoints (results in bold considered significant *p* ≤ 0.017). Δ1 = MCID between baseline and 12 months (early stage). Δ2 = MCID between 12 months and 36 months (early stage). Δ3 = MCID between baseline and 12 months (advanced stage). Δ4 = MCID between 12 months and 36 months (advanced stage). (−) = Negative MCID (QoL deterioration). (+) = Positive MCID (QoL improvement).

**Table 3 cancers-16-02724-t003:** QLQ-H&N35 scores according to cancer stage (early, advanced) at baseline, 12 months and 36 months.

	Baseline P_50_ [P_25_–P_75_]			12 Months P_50_ [P_25_–P_75_]			36 Months P_50_ [P_25_–P_75_]								
	Early (N = 30)	Advanced (N = 42)	*p* (1)	Early (N = 26)	Advanced (N = 40)	*p* (1)	Early (N = 19)	Advanced (N = 32)	*p* (1)	*p* (2)	*p* (3)	Δ1	Δ2	Δ3	Δ4
**QLQ-H&N35 Symptoms scales (100 = unfavourable)**															
Head neck pain Swallowing problems Senses problems Speech Social eating Social contact Less sexuality Teeth Opening mouth Dry mouth Sticky saliva Coughing Feeling ill	20 [16–31] 0 [0–0] 0 [0–0] 0 [0–0] 7.1 [5–10] 0 [0–0] 0 [0–0] 30 [20–35] 0 [0–0] 0 [0–0] 0 [0–0] 0 [0–0] 0 [0–0]	38 [25–50] 14.4 [0–20] 6 [1–9] 21 [18–25] 32 [20–35] 13 [5–18] 25 [20–35] 28 [20–35] 2 [0–5] 0 [0–0] 0 [0–0] 13 [5–18] 0 [0–0]	**0.001** **0.020** **0.002** **0.033** **0.023** **0.009** **0.003** 0.560 **0.001** 0.224 0.331 **0.002** 0.940	22.3 [12–31.6] 0 [0–4.8] 0 [0–4.8] 0 [0–3.5] 14.6 [6.3–25.6] 7.8 [3–10.4] 33.3 [20–53.3] 2.6 [0–9.8] 37.2 [25–47.9] 15.6 [0–20.3] 31.5 [20–53.3] 0 [0–0] 0 [0–0]	33.8 [25–66.6] 18.5 [0–26] 20.4 [0–26] 19.1 [0–36] 34.6 [6.3–50] 15.5 [0–20.6] 72.3 [56–80.8] 37.6 [0–45.7] 71.3 [61–82.3] 33.3 [0–40.2] 69.3 [56–80.8] 0 [0–0] 0 [0–0]	**0.019** **0.022** **0.002** **0.022** **0.014** **0.020** **0.003** **0.015** **0.001** **0.002** **0.033** 0.220 0.190	12.3 [10–16.6] 5.7 [4.6–12.6] 0 [0–0] 9.1 [2.3–11.3] 0 [0–6.3] 0 [0–0] 0 [0–0] 0 [0–0] 0 [0–0] 5.9 [0–12] 10.5 [2.6–12.9] 0 [0–0] 0 [0–0]	25.3 [12–31.6] 15.7 [9.6–23.6] 0 [0–0] 16.9 [9.4–22.5] 14.6 [10–26.3] 5.3 [0–13.6] 26.4 [19–32.5] 0 [0–0] 19.8 [12.5–29] 4.3 [0–13] 9.8 [3.4–13.5] 0 [0–0] 0 [0–0]	**0.019** **0.009** 0.222 **0.004** **0.008** **0.001** **0.019** 0.156 **0.001** 0.220 0.330 0.220 0.190	**0.007** 0.034 0.345 0.858 0.023 **0.008** 0.034 0.019 **0.003** 0.032 0.020 0.456 0.579	0.027 0.030 **0.004** 0.097 **0.003** 0.022 **0.001** **0.008** **0.001** **0.007** 0.033 0.678 0.453	−2.3 0 0 0 −7.5 −7.8 −33.3 +27.4 −37.2 −15.6 −31.5 0 0	+10 −5.7 0 −9.1 +14.6 +7.8 +33.3 +2.6 +37.2 +9.7 +21 0 0	+4.2 −4.1 −14.4 +1.9 −2.6 −2.5 −47.3 −9.6 −69.3 −33.3 −69.3 +13 0	+8.5 +2.8 +20.4 +2.2 +20 +10.2 +45.9 +37.6 +51.5 +29 +59.5 0 0
**QLQ-H&N35 Single items (100 = unfavourable)**															
Painkillers Nutritional supplements Feeding tube Weight loss Weight gain	98 [90–100] 0 [0–0] 0 [0–0] 0 [0–0] 0 [0–0]	98 [90–100] 0 [0–0] 0 [0–0] 95 [80–100] 0 [0–0]	0.489 0.334 0.451 **0.001** 0.220	97 [95–100] 0 [0–0] 0 [0–0] 0 [0–0] 0 [0–0]	98 [95–100] 10 [0–12] 7 [0–12] 96.6 [50–100] 0 [0–0]	0.190 **0.001** **0.013** **0.005** 0.220	0 [0–0] 0 [0–0] 0 [0–0] 0 [0–0] 0 [0–0]	0 [0–0] 0 [0–0] 0 [0–0] 0 [0–0] 0 [0–0]	0.190 0.222 0.222 0.091 0.220	0.034 0.345 0.019 **0.003** 0.032	**0.008** **0.007** **0.008** **0.001** **0.007**	+1 0 +27.4 −37.2 −15.6	+97 0 +2.6 +37.2 +9.7	0 −10 −9.6 −69.3 −33.3	+98 +10 +37.6 +51.5 +29

Abbreviations: P_50_ = Median score. [ ] = Interquartile range. Significance *p*-value (1) = Intergroup comparative analysis at each checkpoint (results in bold considered significant *p* ≤ 0.050). Significance *p*-value (2) = Intragroup comparative analysis according to early stage between the three checkpoints (results in bold considered significant *p* ≤ 0.017). Significance *p*-value (3) = Intragroup comparative analysis according to advanced stage between the three checkpoints (results in bold considered significant *p* ≤ 0.017). Δ1 = MCID between baseline and 12 months (early stage). Δ2 = MCID between 12 months and 36 months (early stage). Δ3 = MCID between baseline and 12 months (advanced stage). Δ4 = MCID between 12 months and 36 months (advanced stage). (−) = Negative MCID (QoL deterioration). (+) = Positive MCID (QoL improvement).

**Table 4 cancers-16-02724-t004:** QLQ-C-30 scores according to cancer treatment (surgery+/−RT-CT, radical RT-CT) at baseline, 12 months and 36 months.

	Baseline P_50_ [P_25_–P_75_]			12 Months P_50_ [P_25_–P_75_]			36 Months P_50_ [P_25_–P_75_]								
	Surgery+/−RT-CT (N = 63)	Definitive CT-RT (N = 9)	*p* (1)	Surgery+/−RT-CT (N = 58)	Definitive CT-RT (N = 8)	*p* (1)	Surgery+/−RT-CT (N = 46)	Definitive CT-RT (N = 5)	*p* (1)	*p* (2)	*p* (3)	Δ1	Δ2	Δ3	Δ4
**QLQ-C-30 Functional scales (100 = favourable)**															
Global health status Physical functioning Role functioning Emotional functioning Cognitive functioning Social functioning	85 [66–90] 96 [90–100] 98 [95–100] 90 [85–100] 90 [85–100] 95 [90–100]	83 [66–90] 95 [90–100] 98 [95–100] 94 [90–100] 85 [80–100] 95 [90–100]	0.140 0.855 0.141 0.759 0.466 0.200	66.6 [56–99.6] 90.4 [66.6–100] 90.4 [86.6–100] 73.3 [56–86.6] 70.5 [86.6–100] 90.1 [85–100]	66.6 [56.6–100] 91.5 [66.6–100] 91.2 [66.6–100] 73.3 [53–86.6] 71.4 [86.6–100] 95.4 [86.6–100]	0.420 0.580 0.586 0.258 0.645 0.562	99.6 [86.6–100] 98 [66.6–100] 98 [86.6–100] 96 [95–100] 98 [86.6–100] 97 [86.6–100]	97.6 [86.6–100] 97 [66.6–100] 97 [66.6–100] 83.3 [86.6–100] 99 [86.6–100] 95 [86.6–100]	**0.001** 0.222 0.800 **0.002** 0.660 0.222	**0.004** 0.043 0.023 0.035 0.039 **0.005**	0.022 0.041 0.025 **0.016** 0.025 0.015	−18.4 −5.6 −7.6 −16.7 −19.5 −4.9	+33 +7.6 +7.6 +22.7 +27.5 +6.9	−16.4 −3.5 −6.8 −20.7 −13.6 +0.4	+31 +5.5 +5.8 +10 +27.6 −0.4
**QLQ-C30 Symptom scales (100 = unfavourable)**															
Fatigue Nausea and vomiting Pain Dyspnoea Insomnia Appetite loss Constipation Diarrhoea Financial difficulties	5 [0–15] 0 [0–0] 33 [15–40] 0 [0–0] 0 [0–0] 0 [0–0] 0 [0–0] 0 [0–0] 0 [0–0]	11 [0–20] 0 [0–0] 30 [15–40] 0 [0–0] 0 [0–0] 0 [0–0] 0 [0–0] 0 [0–0] 0 [0–0]	0.120 0.185 **0.002** 0.354 0.193 0.864 0.537 0.293 0.658	20.3 [0–22.6] 0 [0–0] 25.5 [0–26.6] 0 [0–0] 0 [0–0] 0 [0–0] 0 [0–0] 0 [0–0] 0 [0–0]	20.4 [0–22.6] 0 [0–0] 27.8 [0–33.3] 0 [0–0] 0 [0–0] 0 [0–0] 0 [0–0] 0 [0–0] 0 [0–0]	0.525 0.684 0.522 0.558 0.797 0.866 0.534 0.692 0.658	0 [0–22.6] 0 [0–0] 0 [0–16.6] 0 [0–0] 0 [0–0] 0 [0–0] 0 [0–0] 0 [0–0] 0 [0–0]	0 [0–12.6] 0 [0–0] 0 [0–33.3] 0 [0–0] 0 [0–0] 0 [0–0] 0 [0–0] 0 [0–0] 0 [0–0]	0.520 0.680 0.520 0.550 0.790 0.860 0.530 0.690 0.650	**0.006** 0.673 0.023 0.384 0.547 0.542 0.230 0.235 0.023	**0.005** 0.842 **0.003** 0.348 0.349 0.450 0.348 0.232 **0.001**	−15.3 0 +7.5 0 0 0 0 0 0	+20.3 0 +25.5 0 0 0 0 0 0	−9.4 0 +2.2 0 0 0 0 0 0	+20.4 0 +27.8 0 0 0 0 0 0

Abbreviations: P_50_ = Median score. [ ] = Interquartile range. Significance *p*-value (1) = Intergroup comparative analysis at each checkpoint (results in bold considered significant *p* ≤ 0.050). Significance *p*-value (2) = Intragroup comparative analysis according to surgery+/−RT-CT treatment between the three checkpoints (results in bold considered significant *p* ≤ 0.017). Significance *p*-value (3) = Intragroup comparative analysis according to radical RT-CT treatment between the three checkpoints (results in bold considered significant *p* ≤ 0.017). Δ1 = MCID between baseline and 12 months (surgery+/−RT-CT). Δ2 = MCID between 12 months and 36 months (surgery+/−RT-CT). Δ3 = MCID between baseline and 12 months (radical RT-CT). Δ4 = MCID between 12 months and 36 months (radical RT-CT). (−) = Negative MCID (QoL deterioration). (+) = Positive MCID (QoL improvement).

**Table 5 cancers-16-02724-t005:** QLQ-H&N35 scores according to cancer treatment (surgery+/−RT-CT, radical RT-CT) at baseline, 12 months and 36 months.

	Baseline P_50_ [P_25_–P_75_]			12 Months P_50_ [P_25_–P_75_]			36 Months P_50_ [P_25_–P_75_]								
	Surgery+/−RT-CT (N = 63)	Definitive CT-RT (N = 9)	*p* (1)	Surgery+/−RT-CT (N = 58)	Definitive CT-RT (N = 8)	*p* (1)	Surgery+/−RT-CT (N = 46)	Definitive CT-RT (N = 5)	*p* (1)	*p* (2)	*p* (3)	Δ1	Δ2	Δ3	Δ4
**QLQ-H&N35 Symptoms scales (100 = unfavourable)**															
Head neck pain Swallowing problems Senses problems Speech Social eating Social contact Less sexuality Teeth Opening mouth Dry mouth Sticky saliva Coughing Feeling ill	33 [16–41] 9 [0–25] 0 [0–0] 22 [15–30] 16 [8–30] 0 [0–0] 22 [15–30] 32 [18–40] 0 [0–0] 0 [0–0] 0 [0–0] 0 [0–0] 0 [0–0]	33 [15–50] 8 [0–21] 0 [0–0] 22 [15–30] 30 [12–41] 0 [0–0] 20 [10–25] 30 [14–38] 0 [0–0] 0 [0–0] 0 [0–0] 6 [0–10] 0 [0–0]	0.150 0.515 0.790 0.125 **0.006** 0.165 0.064 0.675 0.182 0.184 0.395 **0.001** 0.283	18.3 [0–33.3] 0 [0–16.6] 0 [0–0] 0 [0–16.6] 16.3 [6.6–25] 0 [0–16.6] 36.3 [0–66.6] 33.3 [0–41.6] 0 [0–33.3] 0 [0–33.3] 0 [0–33.3] 0 [0–0] 0 [0–0]	25 [16–33.3] 3.2 [0–33.3] 0 [0–0] 5 [0–33.3] 22.2 [8.3–33.3] 0 [0–16.6] 33.3 [0–66.6] 33.3 [0–33.3] 0 [0–33.3] 0 [0–6.6] 0 [0–0] 0 [0–33.3] 0 [0–33.3]	0.535 0.622 0.792 0.734 0.627 0.664 0.662 0.674 0.521 0.557 0.752 0.515 0.822	8.3 [0–33.3] 0 [0–16.6] 0 [0–0] 0 [0–16.6] 6.3 [0–25] 0 [0–16.6] 16.3 [0–66.6] 23.3 [0–41.6] 0 [0–33.3] 2.3 [0–33.3] 3.1 [0–33.3] 0 [0–0] 0 [0–0]	5.7 [0–16.6] 3.3 [0–33.3] 0 [0–0] 5.8 [0–16.6] 2.2 [0–16.6] 0 [0–16.6] 33.3 [0–66.6] 23.4 [0–33.3] 0 [0–33.3] 20.4 [0–66.6] 21.8 [0–66.6] 0 [0–33.3] 0 [0–33.3]	0.530 0.620 0.790 0.730 0.620 0.660 0.660 0.670 0.222 **0.001** **0.002** 0.510 0.820	**0.004** 0.034 **0.004** **0.008** 0.023 0.038 **0.014** **0.001** 0.023 **0.002** 0.029 0.456 0.579	0.027 **0.002** 0.023 **0.007** **0.003** **0.002** 0.036 0.033 **0.011** 0.042 0.031 0.678 0.453	+14.7 +9 0 +22 −0.3 0 −14.3 −1.3 0 0 0 0 0	+10 0 0 0 +10 0 +20 +10 0 −2.3 −3.1 0 0	+8 +4.8 0 +17 +7.8 0 −13.3 −3.3 0 0 0 +6 0	+19.3 −0.1 0 −0.8 +20 0 0 +9.9 0 −20.4 −21.8 0 0
**QLQ-H&N35 Single items (100 = unfavourable)**															
Painkillers Nutritional supplements Feeding tube Weight loss Weight gain	99 [95–100] 0 [0–0] 0 [0–0] 99 [95–100] 0 [0–0]	99 [95–100] 0 [0–0] 0 [0–0] 98 [95–100] 0 [0–0]	0.187 0.183 0.397 0.236 0.197	98 [0–100] 0 [0–0] 0 [0–0] 0 [0–100] 0 [0–0]	97 [0–100] 0 [0–0] 0 [0–0] 98 [0–100] 0 [0–0]	0.817 0.814 0.596 **0.012** 0.662	98 [0–100] 0 [0–0] 0 [0–0] 0 [0–16.6] 0 [0–0]	98 [0–100] 0 [0–0] 0 [0–0] 0 [0–0] 0 [0–0]	0.810 0.810 0.590 0.530 0.660	0.852 0.345 0.923 **0.005** 0.347	0.986 0.467 0.456 **0.006** **0.005**	+1 0 0 +99 0	0 0 0 0 0	+2 0 0 0 +99	−1 0 0 +98 0

Abbreviations: P_50_ = Median score. [ ] = Interquartile range. Significance *p*-value (1) = Intergroup comparative analysis at each checkpoint (results in bold considered significant *p* ≤ 0.050). Significance *p*-value (2) = Intragroup comparative analysis according to surgery+/−RT-CT treatment between the three checkpoints (results in bold considered significant *p* ≤ 0.017). Significance *p*-value (3) = Intragroup comparative analysis according to radical RT-CT treatment between the three checkpoints (results in bold considered significant *p* ≤ 0.017). Δ1 = MCID between baseline and 12 months (surgery+/−RT-CT). Δ2 = MCID between 12 months and 36 months (surgery+/−RT-CT). Δ3 = MCID between baseline and 12 months (radical RT-CT). Δ4 = MCID between 12 months and 36 months (radical RT-CT). (−) = Negative MCID (QoL deterioration). (+) = Positive MCID (QoL improvement).

**Table 6 cancers-16-02724-t006:** QLQ-C-30 scores according to type of surgical approach (open surgery, transoral) at baseline, 12 months and 36 months.

	Baseline P_50_ [P_25_–P_75_]			12 Months P_50_ [P_25_-P_75_]			36 Months P_50_ [P_25_–P_75_]								
	Open Surgery (N = 45)	Transoral (N = 18)	*p* (1)	Open Surgery (N = 42)	Transoral (N = 16)	*p* (1)	Open Surgery (N = 35)	Transoral (N = 11)	*p* (1)	*p* (2)	*p* (3)	Δ1	Δ2	Δ3	Δ4
**QLQ-C-30 Functional scales (100 = favourable)**															
Global health status Physical functioning Role functioning Emotional functioning Cognitive functioning Social functioning	58.7 [43–80.5] 80.5 [78–90.1] 83.1 [77–90.7] 73.2 [65–89.7] 83 [80–90.7] 98 [95–99]	49.9 [16–95.8] 96.7 [80–99.0] 98 [95–99] 85 [80–98] 98 [95–99] 98 [95–99]	0.662 0.025 0.856 0.562 0.522 0.862	65 [48–70.5] 60.5 [58–80.1] 53.1 [47–80.7] 53.2 [35–69.7] 63 [50–80.7] 68 [55–89]	54 [36–75.8] 76.7 [60.8–89] 78 [65–89] 55 [40–78] 58 [45–78] 68 [55–89]	0.320 **0.002** **0.006** 0.962 0.520 0.531	89 [87–99] 89 [80–100] 87 [80–100] 92 [90–100] 92 [90–100] 93 [90–100]	93 [87–98] 98 [95–100] 98 [95–100] 95 [90–100] 93 [90–100] 95 [90–100]	0.862 **0.006** **0.028** 0.962 0.217 0.963	**0.002** **0.005** 0.023 **0.008** 0.040 0.033	**0.005** **0.004** **0.006** **0.009** 0.023 **0.004**	+6.3 −20 −30 −20 −20 −30	+24 +28.5 +33.9 +38.8 +29 +25	+4.1 −20 −20 −30 −40 −30	+39 +21.3 +20 +40 +35 +27
**QLQ-C30 Symptom scales (100 = unfavourable)**															
Fatigue Nausea and vomiting Pain Dyspnoea Insomnia Appetite loss Constipation Diarrhoea Financial difficulties	50 [33–66] 0 [0–0] 40 [35–55] 60 [35–85] 10 [0–20] 3 [0–13] 0 [0–0] 0 [0–0] 95 [90–100]	44.8 [33–66.6] 0 [0–0] 42 [33–66] 66 [50–78] 5 [0–18] 5 [0–15] 0 [0–0] 0 [0–0] 95 [90–98.1]	0.354 0.485 0.467 0.385 0.689 0.506 0.439 0.886 0.749	64 [33–66] 0 [0–0] 42 [35–55] 70 [55–89] 15 [0–29] 13 [0–23] 0 [0–0] 0 [0–0] 90 [80–90]	51.9 [33–66.6] 0 [0–0] 42 [33–66] 76 [60–88] 16 [0–28] 15 [0–33] 0 [0–0] 0 [0–0] 90 [70–98.1]	0.863 0.523 0.528 0.963 0.852 0.852 0.863 0.867 0.894	33 [25–55] 0 [0–0] 22 [6–40] 15 [0–26] 15 [0–18] 13 [0–20] 0 [0–0] 0 [0–0] 10 [0–15]	29 [20–60] 0 [0–0] 16 [5–33] 10 [0–35] 16 [0–25] 15 [0–20] 0 [0–0] 0 [0–0] 8 [0–12]	0.962 0.425 0.463 0.521 0.527 0.621 0.125 0.851 0.637	**0.001** 0.041 **0.006** **0.004** **0.001** **0.008** 0.672 0.523 **0.008**	**0.005** 0.020 **0.008** **0.001** **0.005** **0.002** 0.630 0.860 **0.009**	−14 0 −2 −10 −5 −10 0 0 +5	+31 0 +20 +55 0 0 0 0 +80	−7.1 0 0 −10 −11 −10 0 0 +5	+22.9 0 +26 +66 0 0 0 0 +82

Abbreviations: P_50_ = Median score. [ ] = Interquartile range. Significance *p*-value (1) = Intergroup comparative analysis at each checkpoint (results in bold considered significant *p* ≤ 0.050). Significance *p*-value (2) = Intragroup comparative analysis according to open surgery approach between the three checkpoints (results in bold considered significant *p* ≤ 0.017). Significance *p*-value (3) = Intragroup comparative analysis according to transoral approach between the three checkpoints (results in bold considered significant *p* ≤ 0.017). Δ1 = MCID between baseline and 12 months (open surgery). Δ2 = MCID between 12 months and 36 months (open surgery). Δ3 = MCID between baseline and 12 months (transoral). Δ4 = MCID between 12 months and 36 months (transoral). (−) = Negative MCID (QoL deterioration). (+) = Positive MCID (QoL improvement).

**Table 7 cancers-16-02724-t007:** QLQ-H&N35 scores according to type of surgical approach (open surgery, transoral) at baseline, 12 months and 36 months.

	Baseline P_50_ [P_25_–P_75_]			12 Months P_50_ [P_25_–P_75_]			36 Months P_50_ [P_25_–P_75_]								
	Open Surgery (N = 45)	Transoral (N = 18)	*p* (1)	Open Surgery (N = 42)	Transoral (N = 16)	*p* (1)	Open Surgery (N = 35)	Transoral (N = 11)	*p* (1)	*p* (2)	*p* (3)	Δ1	Δ2	Δ3	Δ4
**QLQ-H&N35 Symptoms scales (100 = unfavourable)**															
Head neck pain Swallowing problems Senses problems Speech Social eating Social contact Less sexuality Teeth Opening mouth Dry mouth Sticky saliva Coughing Feeling ill	67.8 [56–79.8] 57 [33–63] 80 [70–87] 99 [95–100] 75.6 [60–81] 77 [55–85] 32 [25–56.9] 35.8 [29–48.9] 65.9 [45–78] 32 [25–57.8] 34.7 [28–41] 68.3 [45–79] 67 [47–79]	75.7 [67–87.8] 70 [59–90] 99 [95–100] 95 [90–100] 92.1 [87–98.4] 78 [55–85] 51.2 [36–67.8] 69.4 [51–74.8] 98 [95–100] 66 [54–78] 62.9 [52–78.9] 98 [95–100] 69 [52–79]	**0.060** **0.001** **0.060** **0.034** **0.012** 0.470 **0.003** **0.001** **0.011** **0.011** **0.019** **0.001** 0.588	77.8 [66–89.8] 63 [43–78] 81 [70–87] 99 [95–100] 76 [60–81] 97 [95–100] 37 [25–56.9] 35.8 [29–48.9] 75.9 [55–87] 41 [25–57.8] 39.7 [28–49] 71.3 [45–79] 67 [47–79]	85.7 [77–97.8] 79 [53–90] 99 [95–100] 95 [90–100] 95 [87–98.4] 98 [95–100] 55 [36–67.8] 69.4 [51–74.8] 98 [95–100] 69 [54–78] 69.8 [52–78.9] 98 [95–100] 69 [52–79]	0.960 **0.007** 0.685 **0.025** **0.009** 0.952 **0.018** **0.040** **0.034** **0.005** **0.007** **0.002** 0.963	20 [0–30] 7 [0–25] 18 [0–33] 15 [0–20] 0 [0–0] 0 [0–0] 0 [0–0] 0 [0–0] 0 [0–0] 11 [0–16] 7 [0–10] 3 [0–8] 0 [0–0]	18 [0–26] 19 [0–33] 20 [0–30] 10 [0–12] 0 [0–0] 0 [0–0] 0 [0–0] 0 [0–0] 0 [0–0] 5 [0–10] 15 [0–20] 16 [0–20] 0 [0–0]	0.921 **0.047** 0.627 **0.009** 0.357 0.967 0.820 0.972 0.966 **0.007** **0.008** **0.010** 0.456	**0.008** **0.006** 0.045 **0.005** **0.001** 0.023 **0.007** **0.010** **0.002** **0.005** **0.005** 0.042 **0.008**	**0.007** 0.025 **0.006** **0.004** **0.012** 0.025 **0.008** **0.004** **0.001** **0.004** 0.023 **0.012** **0.004**	−10 −6 −1 0 −0.4 −20 −5 0 −10 −9 −5 −3 0	+57.8 +56 +63 +84 +76 +97 +37 +35.8 +75.9 +30 +32.7 +68.3 +67	−10 −9 0 0 −2.9 −20 −3.8 0 0 −3 −6.9 0 0	+67.7 +60 +79 +85 +95 +98 +55 +69.4 +98 +64 +54.8 +82 +69
**QLQ-H&N35 Single items (100 = unfavourable)**															
Painkillers Nutritional supplements Feeding tube Weight loss Weight gain	0 [0–0] 99 [95–100] 95 [90–100] 99 [95–100] 98 [95–100]	0 [0–0] 98 [95–100] 96 [95–100] 98 [95–100] 99 [95–100]	0.763 0.749 0.479 0.748 0.579	0 [0–0] 99 [95–100] 95 [90–100] 99 [95–100] 98 [95–100]	0 [0–0] 98 [95–100] 96 [95–100] 98 [95–100] 99 [95–100]	0.224 0.652 0.242 0.920 0.952	0 [0–0] 0 [0–0] 0 [0–0] 0 [0–0] 0 [0–0]	0 [0–0] 0 [0–0] 0 [0–0] 0 [0–0] 0 [0–0]	0.348 0.996 0.332 0.420 0.752	0.122 0.063 **0.004** **0.012** 0.018	0.257 **0.007** **0.008** **0.015** 0.025	0 0 0 0 0	0 +99 +95 +99 +98	0 0 0 0 0	0 +98 +96 +98 +99

Abbreviations: P_50_ = Median score. [ ] = Interquartile range. Significance *p*-value (1) = Intergroup comparative analysis at each checkpoint (results in bold considered significant *p* ≤ 0.050). Significance *p*-value (2) = Intragroup comparative analysis according to open surgery approach between the three checkpoints (results in bold considered significant *p* ≤ 0.017). Significance *p*-value (3) = Intragroup comparative analysis according to transoral approach between the three checkpoints (results in bold considered significant *p* ≤ 0.017). Δ1 = MCID between baseline and 12 months (open surgery). Δ2 = MCID between 12 months and 36 months (open surgery). Δ3 = MCID between baseline and 12 months (transoral). Δ4 = MCID between 12 months and 36 months (transoral). (−) = Negative MCID (QoL deterioration). (+) = Positive MCID (QoL improvement).

## Data Availability

The data presented in this study are available by contacting the corresponding author upon reasonable request.

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
