# Peer review of "Impact of Treatment on Quality of Life in Oropharyngeal Cancer Survivors: A 3-Year Prospective Study"

_cancers, 2024, doi:10.3390/cancers16152724_

Round 1

Reviewer 1 Report

Comments and Suggestions for Authors

This prospespective study aimed to assess the impact on quality of life (QoL) from pretreatment to 3 years after treatment in oropharyngeal carcinoma survivors. QoL was measures with the EORTC QLQ-C30 and EORTC QLQ-HeM35 scales.

However, the methodologically well-conducted study reaches conclusions already consolidated in the international literature. The impossibility of regaining the ability to speak in a short time, for example, leads to

a permanent disability with social isolation and sometimes increasingly serious consequences

exceeding the simple interpersonal relationships established by the patient within the family environment.

The patient with oral cancer, treated and cured, constitutes a presence

socially relevant, much more than what happens for cancer patients cured by others

visceral tumors, as the therapeutic interventions required by the disease, interfere with

multiple physiological functions fundamental for the completion of a complete life

relationship (phonation, swallowing, chewing, aesthetic appearance).

It follows that every therapeutic intervention program must necessarily address,

among others, also the problem of rehabilitating to acceptable levels, the performance status of the patient. The authors used questionnaires for the subjective evaluation of quality of life for numerous factors. But it is possible to integrate related the presence of lateral-cervical or distant metastases affect the quality of life? Does speech therapy, physiotherapy and psychological rehabilitation have an impact on the quality of life?

Comments on the Quality of English Language

/

Author Response

In the following, we respond point by point to the concerns specifically raised by the three reviewers.

REVIEWER #1:

Point #1:

The authors used questionnaires for the subjective evaluation of quality of life for numerous factors. But it is possible to integrate related the presence of lateral-cervical or distant metastases affect the quality of life? Does speech therapy, physiotherapy and psychological rehabilitation have an impact on the quality of life?

Response #1:

Thank you for your comment. The reviewer points out a very important aspect regarding the QoL of patients undergoing oropharyngeal cancer treatment insofar as essential functions are affected and altered. In this sense, functional rehabilitation after oropharyngeal cancer treatment is an important process that can help improve their QoL and allow them to return to normal activities. QoL studies with validated QoL questionnaires, such as the one we conducted in our study, clearly show that treatments for oropharyngeal cancer (surgery, radiation therapy, and chemotherapy) can cause a number of side effects that can affect their ability to speak, eat, swallow, and breathe. There is no doubt that a supportive programme of speech therapy, physiotherapy and psychological rehabilitation, as indicated by the reviewer, can help the patient to manage these side effects and regain function. In Spain, such programmes are carried out at the primary care level outside hospitals, by family doctors, physiotherapists, speech therapists, nutritionists, and social workers. Unfortunately, we do not have data collected on these aspects, as this was not the aim of our study.

On the other hand, we did not detect any cases of distant metastasis in our patients. Although we performed cervical lymph node dissections in 91.7% of patients, we have not specifically studied how this aspect affected QoL. There is a specific QoL questionnaire for this problem (Shoulder Pain and Disability Index - SPADI), as well as other QoL questionnaires aimed more specifically at head and neck cancer symptoms, such as the Voice Related QOL (VRQOL), the MD Anderson Dysphagia Inventory (MDADI), the Swallowing Quality of Life (SWAL QOL) and the MD Anderson Dysphagia Inventory (MDADI), and the Xerostomia-related Quality of Life Scale (XeQoLS). Unfortunately, these analyses were beyond the scope of this study. We cannot provide additional information to the reviewer on these topics.

Reviewer 2 Report

Comments and Suggestions for Authors

The data are longitudinal, yet they are not handled appropriately using repeated measures ANOVA (MANOVA). MANOVA can only study patients with outcomes at all measurement moments, limiting the analysis. A more suitable approach would be Linear Mixed Modelling (LMM). LMM can include all patients in the study and allow examination of changes in outcomes over time, as well as identifying which clinical and demographic factors are related.

Furthermore, LMM can accommodate interaction terms and factors (e.g., treatment modality) to determine if the quality of life trajectory differs by treatment type.

Given the numerous analyses conducted in this study, there is a risk of introducing bias. Therefore, it is essential to apply the Bonferroni correction to adjust for multiple comparisons.

Author Response

REVIEWER #2:

Point #2:

Given the numerous analyses conducted in this study, there is a risk of introducing bias. Therefore, it is essential to apply the Bonferroni correction to adjust for multiple comparisons.

Response #2:

Thank you for pointing this out, we agree with this comment. As the reviewer suggests, we collected intragroup QoL scores with the EORTC questionnaire at three checkpoints (pre-treatment, 1 year and 3 years post-treatment) and therefore the Bonferroni correction can be applied to adjust p-values and control for the type I error rate (false positives) in multiple comparisons. Consequently, we have revised the Bonferroni correction to an adjusted significance level (p≤0.0167) in the intragroup analysis between the three checkpoints. In making these changes, we note that this correction has had a modest impact on our results, but this adjustment ensures the robustness of our findings amidst the numerous comparisons made. Although we have found that the essential results have not altered much after applying the Bonferroni correction, we detect that there is a change in some items, as this correction is more conservative and reduces the likelihood of false positive results. In the intragroup analysis at the three checkpoints, the number of items with significant differences was reduced. The analysis becomes more conservative and focusses on items where the differences are clearer and stronger.

Consequently, in the new version we have introduced the relevant changes in the Tables and text (marked in red). It should be noted that, for better visualisation by the reader and to facilitate the editing of the Tables, we have changed the presentation of the data, as instead of the 3 Tables we had in the first version, we have divided them into 6 Tables. In this way, we present the scores of the QLQ-C-30 and the QLQ-H&N35 separately. At the end of each Table, we define and clarify the calculations that we consider most clarifying for the reader. Thank you once again for your valuable comments. In the new version, we incorporate all these changes following your recommendations.

Reviewer 3 Report

Comments and Suggestions for Authors

Ths study aimed to assess the impact on quality of life from pretreatment to 3 years after treatment in oropharyngeal carcinoma (OPC) survivors.

The article is interesting,rather well written easy to read. Apparently the first of this type in Spain.

The limitations presented by the study are mentioned.

Here are my comments:

- The abstract should involve a conclusion drawn from the results and applicable broadly

- The safety margins realized in surgery should be stated

- Alcohol and tobacco consumption should be provided more in detail (g/day, pack /d) ansd may be analyzed for differences

- The major point of criticism is that the legend of the clinical photos provided does not match the location of the tumor (one example base of the tongue etc).

Finally, the statement "long term" seems excessive as it is 3 years, it would have interesting to assess the 5 years QoL.

Author Response

REVIEWER #3:

Point #3:

The abstract should involve a conclusion drawn from the results and applicable broadly.

Response #3:

Thank you. Following your recommendations, we have added a new sentence to the Conclusions of the Abstract (marked in red). However, we cannot expand it further to avoid exceeding the number of words allowed in this section.

Point #4:

The safety margins realized in surgery should be stated. Alcohol and tobacco consumption should be provided more in detail (g/day, pack /d) and may be analysed for differences.

Response #4:

We thank the reviewer for this comment. In the Results section, we briefly describe the safety margins we obtained after surgery (marked in red). In addition, we provide new data on tobacco and alcohol consumption in our patients (marked in red). Our QoL study focused on calculating clinically significant changes (≥10) in QoL taking into account tumour stage and treatment modality. Therefore, we did not collect additional details on tobacco consumption in packs/day or alcohol consumption in g/day. After completion of patient data collection, we are currently unable to analyse statistical differences in the influence of these variables on QoL because this was not the aim of our study. We hope that this is not a serious limitation for the reviewer.

Point #5:

The major point of criticism is that the legend of the clinical photos provided does not match the location of the tumor (one example base of the tongue etc).

Response #5:

Thank you for this observation. To classify the different tumour locations in the oropharynx, in our study, we used the International Classification of Diseases (ICD) coding systems (ICD-9 and ICD-10) to categorise the tumour locations. These locations included the base of the tongue (C01), lingual tonsil (C02.4), soft palate (C05.1) and tonsil/oropharynx (146, C09-91 10). In particular, we consider that the location that the reviewer notes as the "base of the tongue" corresponds anatomically to the posterior third of the tongue, behind the circumvallate papillae that extend behind the cecum foramen and the lingual V-shape. We believe this can be seen in figures 1D and 2. However, the inclusion of these Figures of some clinical cases contained in Figures 1, 2, 3 and 4 are not essential to show our QoL results. If the reviewer feels that they confuse the reader, we are open to removing these Figures from our article in the belief that they do not detract from the more relevant information in the EORTC questionnaire.

 Point #6:

Finally, the statement "long term" seems excessive as it is 3 years, it would have interesting to assess the 5 years QoL.

Response #6:

Thank you for pointing this out. In the context of cancer patient studies, the definition of "long-term" does not have a single universally accepted threshold, as it is largely dependent on the objective of the study. In long-term survival/recurrence studies, a follow-up period of 5 years should be assessed. However, if the purpose of the study is to assess long-term side effects of treatment or patient QoL, a follow-up period of 2-3 years may be sufficient. After 3 years there is no change in QoL, only in survival. Most longitudinal studies on QoL in the literature report results at 12 months and long-term QoL studies report results at 24 or 36 months. From 36 months onwards, the results obtained will be very similar, as from 24/36 months the patient QoL in relation to the treatments is considered stabilised and will not change.

Round 2

Reviewer 3 Report

Comments and Suggestions for Authors

Dear Authors,

Thank you for having modified and /or explained the points raised.

Best wishes.